

# 2-D finite displacements and finite strain from PIV analysis of plane-strain tectonic analogue models

David Boutelier[1], Christoph Schrank[2], and Klaus Regenauer-Lieb[3]

[1]School of Life and Environmental Sciences, UON, Newcastle
[2]School of Earth, Environmental and Biological Sciences, QUT, Brisbane
[3]School of Minerals and Energy Resources Engineering, UNSW, Sydney

**Correspondence:** D. Boutelier (david.boutelier@newcastle.edu.au)

**Abstract.** Image correlation techniques have provided new ways to analyze the distribution in space and time of deformation in analogue models of tectonics. Here we demonstrate how the correlation of successive time-lapse images of a deforming model allows not only to evaluate the components of the strain-rate tensor at any time in the model but also calculate the finite displacements and finite strain tensor. We illustrate, using synthetic images, the ability of the algorithm to produce maps of the velocity gradients, small-strain tensor components, but also incremental or instantaneous principal strains and maximum shear. The incremental displacements can then summed up using a Eulerian or a Lagrangian summation, and the components of the 2-D finite strain tensor can be calculated together with the finite principal strain and maximum finite shear. We benchmark the measures of finite displacements using specific synthetic tests for each summation mode. The deformation gradient tensor is calculated from the deformed state, and decomposed into the finite rigid-body rotation and left or right finite stretch tensors, allowing the deformation ellipsoids to be drawn. The finite strain has long been the only quantified measure of strain in analogue models. The presented software package allows producing these finite strain measures while also accessing incremental measures of strain. The more complete characterization of the deformation of tectonic analogue models will facilitate the comparison with numerical simulations and geological data, and help produce conceptual mechanical models.

## 1 Introduction

The concept of physical similarity rests on the idea that multiple physical systems may share the same underpinning physical laws and therefore one can draw inferences from observations in any of the similar systems (Sterrett, 2009, 2017a, b, and reference therein).

In Earth sciences, scaled models have been employed for over a century to test hypotheses on the driving mechanisms of tectonic processes derived from, and constrained by, a variety of geological and geophysical data (e.g. Koyi, 1997; Ranalli, 2001; Graveleau et al., 2012, and reference therein). Ideally, the scaled model passes through an evolution, which simulates that of the original, but on a much more convenient time scale (e.g. Buckingham, 1914; Hubbert, 1937; Ramberg, 1967; Shemenda,





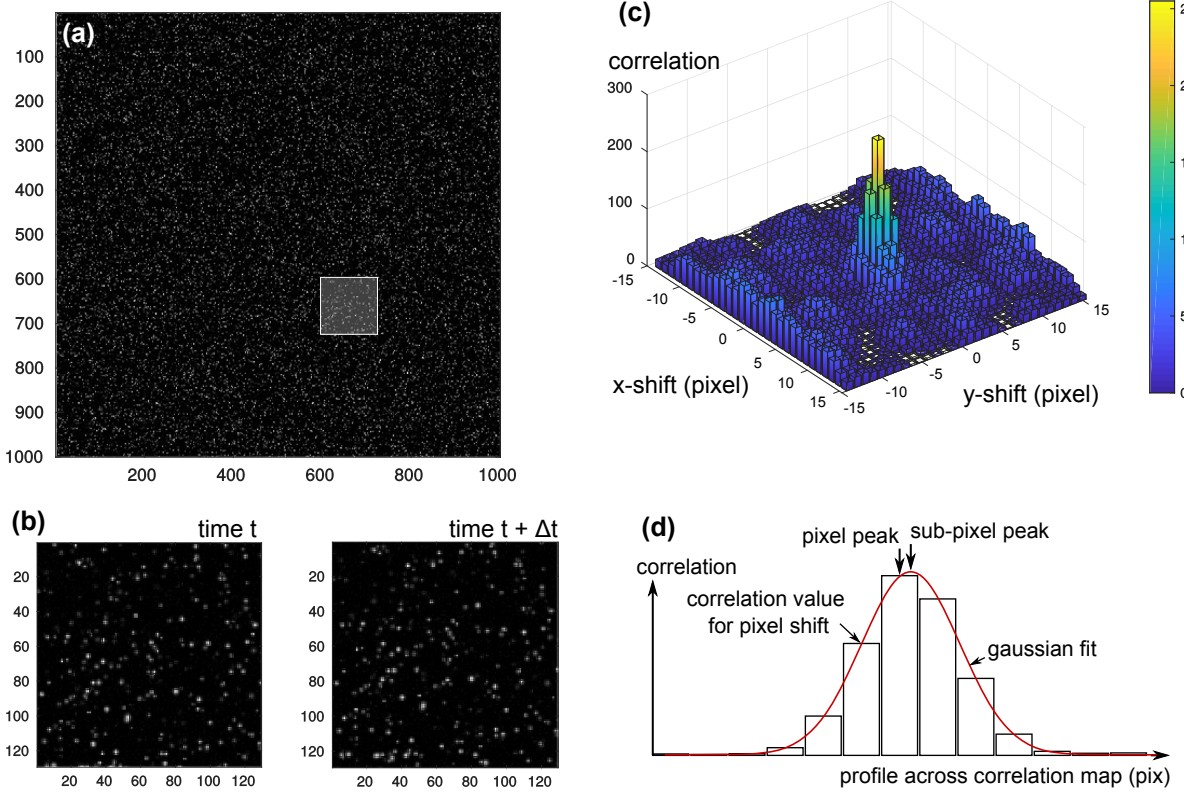

**Figure 1.** Principles of PIV. **(a)**: pre-processed image at time $t$ with illuminated particles; **(b)**: images at time $t$ and $t + \Delta t$ are split into small interrogation windows (here 128 pixels) and the interrogation windows are cross-correlated; **(c)** correlation yields a map, indicating how well the interrogation windows correlate with each other when shifted along x and y axes of the image space (here interrogation windows were 32 pixels wide); **(d)**: the peak of the correlation map is precisely (i.e. sub-pixel) located using a Gaussian fit.

1994; Merle, 2015). The physical process can thus be monitored more precisely in time and space in the scaled analogue model than in nature, which facilitates its evaluation and interpretation.

In tectonic analogue modelling, the monitoring of model kinematics - local velocities and displacements - provided particularly useful insights (e.g. Cruden et al., 2006; Boutelier and Cruden, 2008; Duarte et al., 2013). In the last two decades

5    or so, one specific monitoring method for model velocities was established as the new experimenter's gold standard: Particle Imaging Velocimetry (PIV) (e.g. Adam et al., 2005; Schrank et al., 2008; Rosenau et al., 2009; Adam et al., 2013; Saumur et al., 2015; Boutelier, 2016; Kavanagh et al., 2015, 2017; Zwaan et al., 2018; Dooley et al., 2018; Rudolf et al., 2019). PIV was developed in the fluid-dynamics community to track fluid flow (Adrian, 2005, and reference therein). Fluids were seeded with passive tracers whose motion was observed with time-lapse photographs. The flow field was then obtained through an

10    image correlation technique (e.g. Adrian, 1991, 2005; Raffel et al., 2007). In tectonic laboratory experiments, PIV is used to



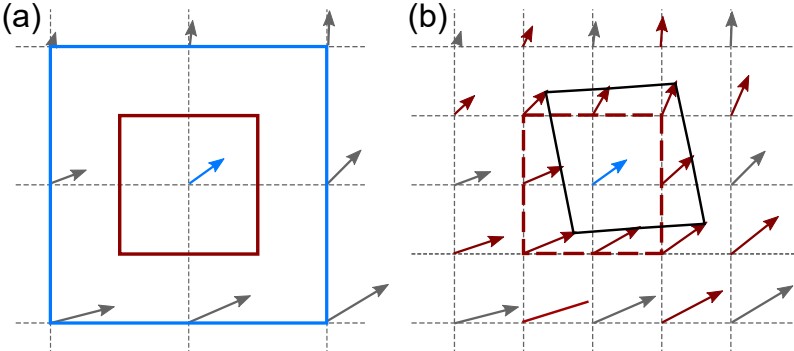

**Figure 2.** Multipass PIV with window deformation. **(a)**: First pass (blue area) with 50% overlap, generates the grey vectors as well as blue vector in center; **(b)**: Second step with smaller interrogation window (red area in panel a). The results of the first pass is interpolated linearly or using a spline function, onto the vector grid of the second pass. The interpolated vectors (red vectors) are employed to shift and deform the interrogation prior to correlation. The correlation is then employed to correct the interpolated vectors.

monitor the displacements/velocities of markers in or on the model to compute strain or strain-rate components, thus revealing where, how, and when deformation occurs in the model.

Because commercial PIV systems are primarily designed for the quantification of rapid fluid flow, these systems usually employ expensive, high-sensitivity but relatively low-resolution digital cameras (<10 Mpixels). The high sensitivity is required to capture the rapid motion of the tracers in the fast-moving fluid at high frequency. The low resolution is a consequence of the required high frequency because it remains technically challenging to move or store a large number of large images at high rates. However, many analogue modeling experiments of tectonics are conducted at very low rates (i.e. $<1\,\mathrm{mm\,s^{-1}}$) (e.g. Corti et al., 2003) or are not rate-sensitive at all, and therefore the rate can be set as low as desired (e.g. Hoth et al., 2007). Thus, lower sampling rates can be used, allowing the use of less expensive, higher resolution (up to 50 Mpixels) consumer-grade digital cameras (DSLR) to capture sufficient light during the long exposure time. For the purpose of facilitating the PIV analysis of slow analogue models of tectonics, we developed a MATLAB GUI allowing, within a single interface, to calibrate and correct the views of the model, correlate the successive images and produce image outputs with incremental displacements, and their spatial derivatives (Boutelier, 2016). Indeed, a number of recently published studies using PIV in the tectonics community report incremental (ie. instantaneous) measures of displacement (e.g. Schmatz et al., 2010; van Gent et al., 2010; Marshak et al., 2019). However, these measures are more akin to GPS-derived velocity fields and calculated strain-rate maps (e.g. Allmendinger et al., 2005; Gupta et al., 2015; Chousianitis et al., 2015; Gunawan and Widiyantoro, 2019) than measures of finite deformation recorded in the rocks (Hindle et al., 2002; McQuarrie, 2002; Oncken et al., 2006; Arriagada et al., 2008). For the comparison of laboratory models to field data related to long-term tectonic processes, finite-strain measures are more useful. Here, we update the previously published PIV package through the implementation of the Eulerian and Lagrangian sums of displacements and components of the 2-D finite strain tensor. We detail and benchmark the technique using synthetic images.





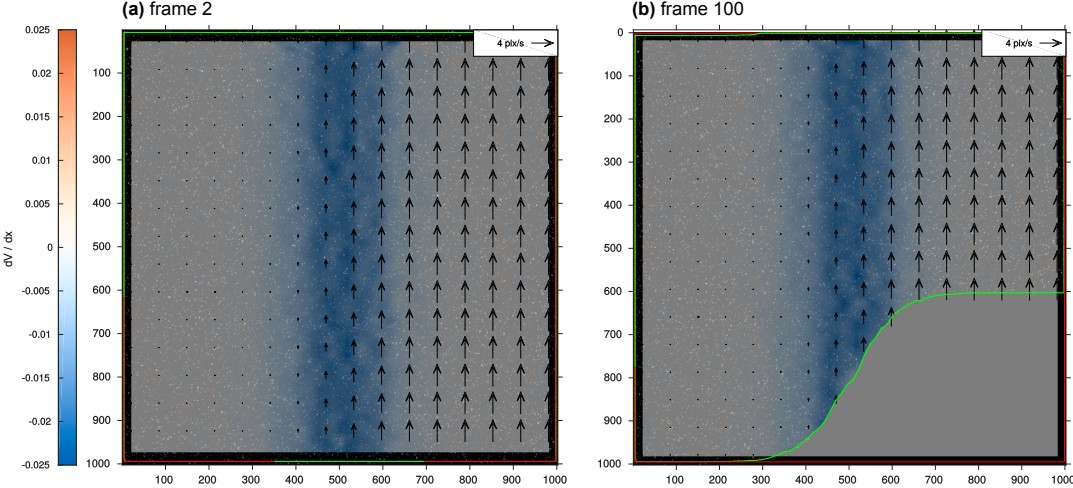

**Figure 3.** Synthetic static shear zone test at frame 2 (a) and 100 (b). Width of the shear zone: $W_{sz} = 512$ pixels, Y-component of the velocity vector: $v = 4$ pixels. Red is the region of interest (ROI), green is the deformed mask.

## 2 Particle Imaging Velocimetry

### 2.1 Principles

In fluid dynamics experiments monitored with PIV, a plane within the fluid is illuminated using a laser sheet (e.g. Adrian, 2005; Raffel et al., 2007). The light scattered by particles in the fluid is recorded on two successive images. Each image is then divided
5  into small interrogation areas, and the local displacement vector between the two images is determined for each interrogation area by cross-correlation (e.g. Adrian, 1991, 2005; Raffel et al., 2007). The algorithm calculates the shift, or displacement vector, for which each interrogation window best matches the next image (Fig. 1). Given the time interval between the two images and the image scaling, the local flow velocity vector can be deduced.

### 2.2 Multipass with window deformation

10  Our software package now allows performing the correlation in multiple successive steps (passes) with decreasing window size and with window deformation. A first pass is performed, where the image pair is analyzed using a relatively large interrogation area and overlap of 50%. The resulting grid of vectors is interpolated to the smaller spatial resolution of the second pass. This interpolated vector field is then employed to pre-shift and pre-deform the interrogation area before the correlation (Fig. 2). The result of the second correlation is thus a correction of the interpolated first pass. A third and fourth pass can be added following
15  the same principle (Scarano, 2002).



**Figure 4.** PIV results for the narrow static shear zone synthetic test. Shear zone width is 64 pixels, y-axis velocity component of the right-hand side is 2 pixels per frame. **(a)** single pass interrogation area width: $w = 128$ pixels; **(b)** single pass with $w = 64$ pixels; **(c)** $w = 32$ pixels; **(d)** multipass with $w = 128$ followed by $w = 64$ pixels; **(e)** $w = 64$ followed by $w = 32$ pixels ; **(f)** $w = 32$ followed by $w = 16$ pixels; **(g)** $w = 64$ followed by $w = 32$ and $w = 16$ pixels; **(h)** $w = 128$ followed by $w = 64$, $w = 32$, and $w = 16$ pixels.



### 2.2.1 Multipass performance

To test how well the new window deformation algorithm captures narrow shear zones, we created a series of synthetic images with a simple static shear zone (see synthetic shear zone test in Boutelier, 2016). The synthetic images are $2000 \times 2000$ pixels and include 240,000 particles with a diameter of $3 \pm 1$ pixels, corresponding to an area proportion of ca. 60%. The right-hand

side of the shear zone is moving upward at a constant velocity, while the left-hand side is static, with a shear zone in between (Fig. 3). The velocity distribution within the shear zone follows the error function. Therefore, the strain-rate profile across the shear zone is a bell-shaped curve. The imposed and measured vertical velocity are compared, and the error is quantified.

For a narrow shear zone with a width of 64 pixels, a single pass with an interrogation area width of 128 pixels and overlap of 50% captures the rigid-body translations very well in the sense that there is very little noise. The magnitude of the velocity

vector is very accurate and the measure is spatially very consistent. However, it fails to capture the width of the shear zone because the vectors are clearly too far apart (Fig. 4a): the effective grid spacing is of the same order as the shear zone width.

A single pass with an interrogation window of 64 or 32 pixels length with overlap of 50% captures the width of the shear zone more accurately, but an edge length of 64 pixels is still too large a window size, while 32 pixels edge length yield spurious variations in the magnitude of the velocity vectors where rigid-body translation is expected (Fig. 4b-c).

A multipass approach with a first pass with an interrogation window size of 64 pixels, overlap of 50% followed by a second pass with a window size of 32 pixels, overlap 50% and linear window deformation captures the shear zone with the precision of the small interrogation window while significantly reducing the noise outside the shear zone (Fig. 4e).

A three-passes approach with a first window of 64 pixels, a second window or 32 pixels and finally a third window of 16 pixels yields a strain profile which captures the shear zone very accurately and does not display severe noise outside the

shear zone (Fig. 4g). A four-passes approach with interrogation windows of 128, 64, 32 and 16 pixels does further reduce the standard deviation of the residuals but the improvement is too small to be noticed, while the processing time is increased by another 20%.

The same data set was processed with a single pass approach with an interrogation window of 64 pixels and an overlap of 75%. This provided a vector grid with a resolution of 16 pixels (i.e. a vector every 16 pixels), similar to the grids produced

by either a single pass with an interrogation area of 32 pixels and overlap of 50%, or any multipass approach with a final pass with an interrogation area of 32 pixels and overlap of 50%. Using the larger overlap (75%), the error on the velocity was 0.1045 pixels. This improved over the single pass using an interrogation window of 32 pixels and an overlap of 50% which gives an error of 0.124 pixels (Fig. 4c). However, the multipass approach with an interrogation window decreasing from 64 to 32 (2 passes, Fig. 4e), or from 128 to 32 (3 passes) provided errors of 0.0632 and 0.0622 pixels respectively. We therefore

confirm that the multipass approach with window deformation does provide a more accurate capture of the velocity distribution than a single pass with large overlap (Scarano, 2002).

In order to capture the strain distribution within a shear zone, it is necessary to have a vector grid spacing that is at least 1/8th of the shear zone width. However, for narrow shear zones this may require the use a small interrogation window which can lead to an increased error because a small window has a less unique distribution of values than a large one. We note that the



use of the multipass approach with window deformation does indeed facilitate monitoring narrow shear zones since the small window is employed last and informed by the the previous passes. We therefore recommend a strategy that employs at least 2 or 3 passes, finishes with an interrogation area sufficiently small that the grid spacing can capture the strain distribution in the monitored shear zones.

However, this strategy may need to be modified depending on the scale of the image, particle size and distribution density. There is a minimum width of the interrogation area associated with the physical size of the particles and image scale. It is obvious that to monitor narrow shear zone, very small particles (several times smaller than the shear zone width), and high image scale (many pixels per physical unit of distance) are better than large particle and low image scale. The minimum width of the interrogation area may also depend on the particle distribution density. If the density is too low, a small interrogation area

may not have sufficiently unique distribution of values to be correlated, or it can be correlated anywhere (no correlation peak). If that is the case the interrogation area should be increased to include more particles and obtain a more unique distribution of values required for correlation.

## 2.3    Velocity gradient tensor

A common practice to evaluate where deformation (or rigid-body rotation) is produced in a model is to calculate the spatial

derivatives of the velocity vectors. The gradients are calculated for each component using the central difference for interior data points. For example, for the $u$ component of the velocity vectors:

$$\frac{\Delta u(i,j)}{\Delta x} = \frac{u(i+1,j) - u(i-1,j)}{2\Delta x} \tag{1}$$

where $i$ and $j$ are the indexes of the velocity grids corresponding to axes $x$ and $y$, respectively, and $\Delta x$ is the homogeneous spacing of the grid along the x-axis. Values along the edges of the domain are calculated with a single-sided difference and

therefore the produced grids of velocity gradients have the same dimension as the velocity grids. The velocity gradients are particularly useful for the analysis of models in which deformation is not confined to the imaged plane. Figure 3 shows an example of a velocity gradient plotted with the velocity vectors for a synthetic static shear zone test. Also displayed are the extend of the region of interest (red rectangle in Fig. 3) and the mask advected with the vectors (green outline in Fig. 3) at the beginning (frame 2) and the end (frame 100) of the test.

## 25   2.4    2D strain-rate tensor

The PIV-derived velocity gradient tensor $\mathbf{D}$ composed of the spatial derivatives of the velocity can be split into symmetric ($\mathbf{S}$) and anti-symmetric ($\mathbf{A}$) parts:

$$\mathbf{D} = \begin{bmatrix} \Delta u/\Delta x & \Delta u/\Delta y \\ \Delta v/\Delta x & \Delta v/\Delta y \end{bmatrix} \tag{2}$$

and

$$\mathbf{D} = \mathbf{S} + \mathbf{A} \tag{3}$$

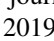
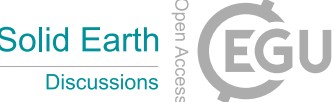


The tensor $\mathbf{S}$ is the 2D small strain rate tensor. $\mathbf{S}$ is derived from $\mathbf{D}$ :

$$\mathbf{S} = \frac{1}{2}\left(\mathbf{D} + \mathbf{D}^T\right) \tag{4}$$

or

$$\mathbf{S} = \begin{bmatrix} \Delta u/\Delta x & \frac{1}{2}\left(\Delta u/\Delta y + \Delta v/\Delta x\right) \\ \frac{1}{2}\left(\Delta v/\Delta x + \Delta u/\Delta y\right) & \Delta v/\Delta y \end{bmatrix} \tag{5}$$

Its components are often noted $\dot{e}_{xx}$, $\dot{e}_{xy}$, and $\dot{e}_{yy}$. Each component of the strain rate tensor can be plotted as a scalar together with the velocity vectors. We note that since the velocities are obtained by dividing the incremental displacements obtained from the correlation by the time interval between the two correlated images, the 2D incremental strain tensor can be obtained from the 2D strain-rate tensor by multiplying it by the time interval. Figure 5 shows $\dot{e}_{xx}$ and $\dot{e}_{xy}$, components of the 2D small strain rate plotted as scalars together with the velocity vectors in the synthetic vortex test. Synthetic images are created where

particles are advected according to velocity distribution corresponding to a vortex with a centre at the centre of the image and a radius of 400 pixels. Within the vortex the magnitude of the velocity vector increases linearly with the distance from the centre. Outside the vortex, the velocity decreases non-linearly with the inverse of the distance to the centre. The positions of the areas with positive and negative values, the symmetries, and reached peak magnitudes fit very well the predicted values from the imposed velocity field, although some anomalies are produced along the left and top edges of the image space.

**2.5   2D rotation tensor**

The anti-symmetric part of the velocity gradient tensor is :

$$\mathbf{A} = \frac{1}{2}\left(\mathbf{D} - \mathbf{D}^T\right) \tag{6}$$

or

$$\mathbf{A} = \begin{bmatrix} 0 & \frac{1}{2}\left(\Delta u/\Delta y - \Delta v/\Delta x\right) \\ \frac{1}{2}\left(\Delta v/\Delta x - \Delta u/\Delta y\right) & 0 \end{bmatrix} \tag{7}$$

$\mathbf{A}$ can be written as:

$$\mathbf{A} = \omega \begin{bmatrix} 0 & -1 \\ 1 & 0 \end{bmatrix} \tag{8}$$

with

$$\omega = \frac{1}{2}\left(\frac{\Delta v}{\Delta x} - \frac{\Delta u}{\Delta y}\right) \tag{9}$$

The anti-symmetric tensor can be characterised by one single parameter, the vorticity $\omega$, which can be plotted as a scalar value

together with the velocity vectors (Fig. 5c). Our vortex synthetic test shows a constant and homogeneous value of the vorticity within the vortex, as predicted from the imposed velocity distribution (Fig. 5c).





## 2.6 2D Invariants of deformation

The components of a tensor change under a change of coordinate system associated with a rotation. However, certain tensor properties remain constant regardless of the choice of coordinate system and are thus called invariants. In 2-D, there are only two invariants. The first invariant relates to the area (volume in 3-D) strain or size change:

$$I_{1e} = tr(\mathbf{S}) = e_{xx} + e_{yy} \tag{10}$$

while the second relates to the deviatoric strain or shape change:

$$I_{2e} = det(\mathbf{S}) = e_{xx}e_{yy} - e_{xy}^2 \tag{11}$$

Our software package now allows plotting these scalar invariants together with the vectors (Fig. 5d). Our synthetic vortex test shows that plotting the second invariant of the small strain tensor allows highlighting the area where the greatest changes in velocity magnitude are, regardless of the orientation of the velocity vector (Fig. 5d). It is an objective measure of deviatoric strain in the model.

## 2.7 Principal strain

The principal strains are two mutually perpendicular directions along which there are no shear strain, only stretch. The stretches in the directions of the principal strains are principal stretches $S_{max}$ and $S_{min}$, and the extensions in these directions are principal extensions $e_{max} = S_{max} - 1$ and $e_{min} = S_{min} - 1$.

The orientations and magnitudes of the incremental principal strains, stretches or extensions can be obtained using the transformation equations describing the changes in the small strain tensor associated with a rotation of the reference frame. In 2-D, the transformation equations are:

$$e'_{xx} = e_{xx}\cos^2\theta + e_{yy}\sin^2\theta + 2e_{xy}\sin\theta\cos\theta \tag{12}$$

$$e'_{yy} = e_{xx}\sin^2\theta + e_{yy}\cos^2\theta - 2e_{xy}\sin\theta\cos\theta \tag{13}$$

$$e'_{xy} = (e_{yy} - e_{xx})\sin\theta\cos\theta + e_{xy}(\cos^2\theta - \sin^2\theta) \tag{14}$$

with $e'_{xx}$, $e'_{yy}$ and $e'_{xy}$ the components of the small strain tensor with the reference frame rotated by the angle $\theta$.

The principal strain orientation, called the principal strain angle $\theta_p$, can be calculated by setting the rotated shear strain to 0 in Eq. 14 and solving for $\theta$:

$$\theta_p = \frac{1}{2}\tan^{-1}\left(\frac{2e_{xy}}{e_{xx} - e_{yy}}\right) \tag{15}$$

Inserting $\theta_p$ back into the equations for the normal strains (Eqs. 12 and 13) gives the principal strain values:

$$e_{max} = \frac{e_{xx} + e_{yy}}{2} + \sqrt{\left(\frac{e_{xx} - e_{yy}}{2}\right)^2 + e_{xy}^2} \tag{16}$$

$$e_{min} = \frac{e_{xx} + e_{yy}}{2} - \sqrt{\left(\frac{e_{xx} - e_{yy}}{2}\right)^2 + e_{xy}^2} \tag{17}$$

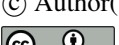



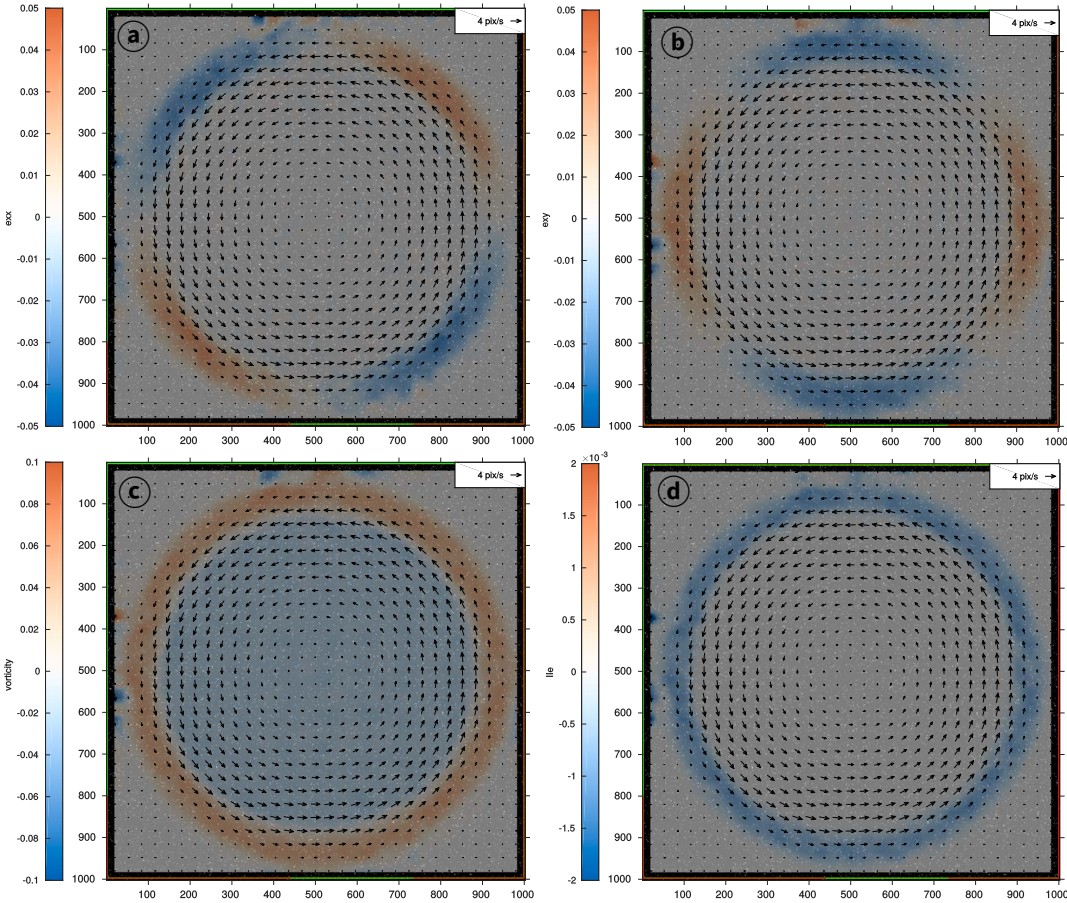

**Figure 5.** PIV results of the vortex synthetic test illustrating some of the scalar parameters that can be derived from the velocity field. **(a)** $e_{xx}$; **(b)** $e_{xy}$; **(c)** vorticity; **(d)** second invariant of the small strain tensor $I_{2e}$.

The maximum amount of shear is calculated from the principal strains:

$$\gamma_{max} = e_{max} - e_{min} \tag{18}$$

The maximum shear always occurs in a coordinate system that is rotated $45°$ from the principal coordinate system.

Once the magnitudes and orientation of the principal strains are calculated, the incremental or instantaneous deformation
ellipse can be plotted for each point of our PIV vector grids. In figure 6, the velocity vectors of a synthetic shear zone test are
plotted and the maximum shear is plotted as a scalar (Fig. 6a). The velocity vectors all possess the same component $u$ along the
x-axis, but the component $v$ along the y-axis changes across a shear zone parallel to the y-axis and located at $x = 580$ pixels
at frame 25. Consequently the measure of $\gamma_{max}$ plotted as a scalar value shows the position of the shear zone, and distribution
of shear within the shear zone (Fig. 6a). Then the principal strains are calculated and the instantaneous principal extensions
are plotted as divergent (negative extension) or convergent (positive extension) couple of arrows in a subset of the region of





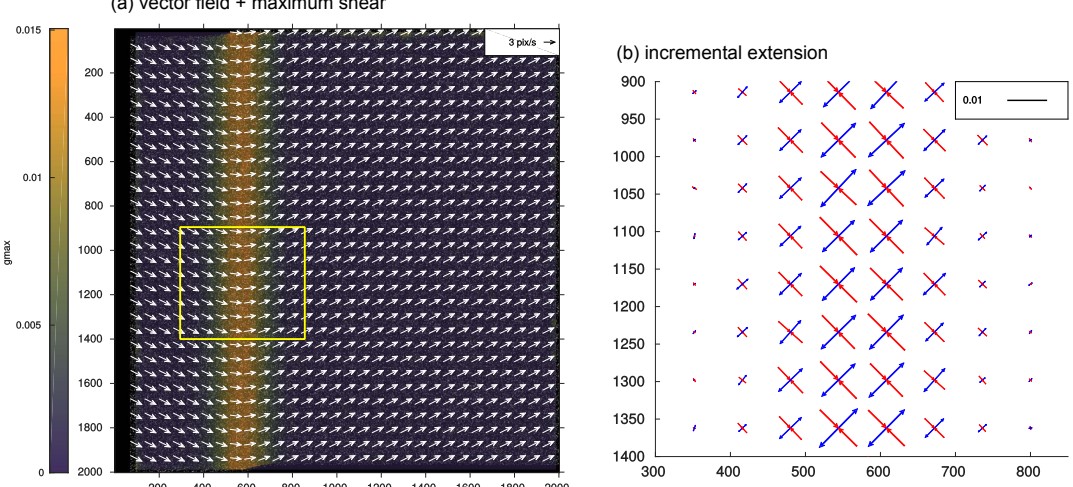

**Figure 6.** Incremental extensions from PIV. **(a)**: synthetic PIV test with vertical shear zone. Horizontal velocity is homogeneous, but vertical velocity changes sign across a vertical shear at X = 580 pixels at frame 25. The shear is illustrated with $\gamma_{max}$, the difference between the principal strains. **(b)** Principal extensions $e_{max}$ and $e_{min}$ across the shear zone. Deformation ellipsoids could have been drawn as well, but the incremental deformation between successive images is very small and the incremental deformation ellipsoids would be very close to circular.

interest. A deformation ellipsoid could have been drawn as well, but the incremental deformation between successive images is very small and the incremental deformation ellipsoids would be very close to circular. It can be seen however, that a unit circle in the shear zone becomes an ellipse with a minor axis corresponding to the converging arrows, and long axis corresponding to the diverging arrows. The orientation of the ellipse associated with the incremental extension does fit the imposed movement

5 (see velocity vectors in Fig. 6a).

## 3 Finite displacements and strain

The calculation of finite deformation is very useful for the analysis and interpretation of tectonic models because it is finite deformation that is most commonly and easily observed in the field. In order to calculate the finite deformation in the model, one may first calculate the sum of incremental displacements. This sum can be performed using a Eulerian or Lagrangian

10 approach.

### 3.1 Eulerian sum of displacements

The Eulerian sum is straightforward when the employed ROI and mask are static since the displacements can be summed up at points which are fixed in space throughout the entire recording. When the mask and ROI deform with the flow, the positions of



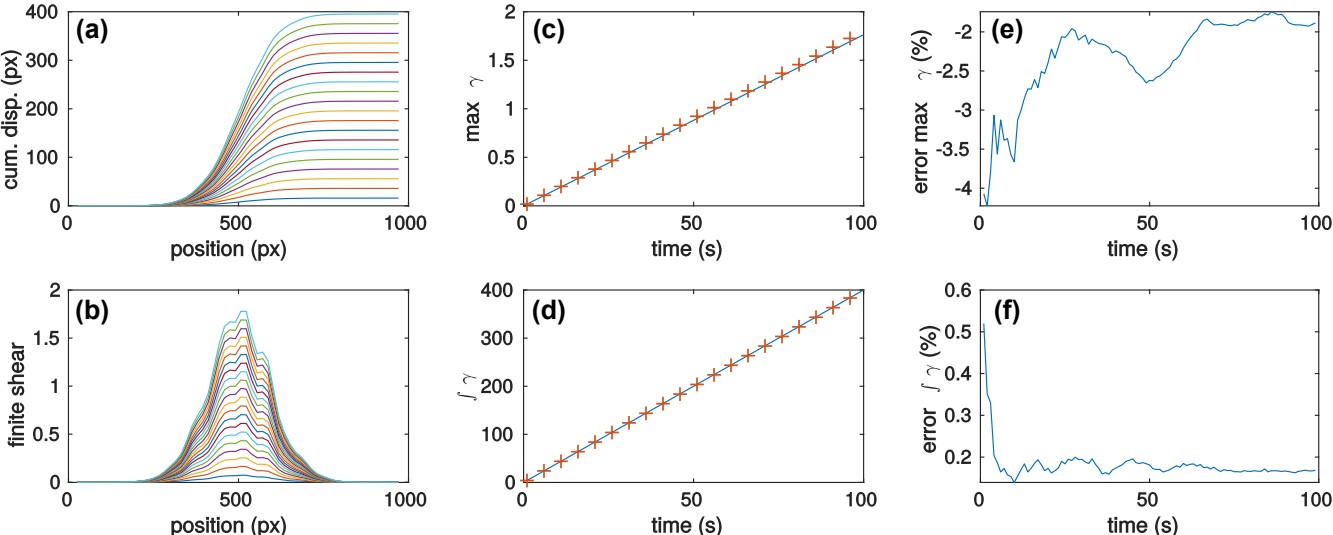

**Figure 7.** Benchmark of the Eulerian sum of displacements using a synthetic static shear zone. **(a)**: cumulative displacement profiles across the static shear zone (profile plotted every 5 frames only for clarity); **(b)**: gradients of the cumulative displacement profiles from panel a; **(c)**: evolution of the peak of the gradient of the cumulative displacements (i.e. max shear, $\gamma$) versus time (assuming 1 s between frames in synthetic tests) measured from PIV (red crosses) versus the solution (blue line) every 5 frames; **(d)**: integral of the shear profile versus time measured from PIV (red crosses) and solution (blue line) every 5 frames; **(e)**: error on the peak cumulative shear strain; **(f)**: error on the integrated cumulative shear strain.

the points were incremental displacements are known will change, and the new points need to be interpolated. For simplicity we perform the Eulerian sum only within a fixed mask.

The components $U_E, V_E$ of the finite displacement at coordinates $x, y$ and time $t_e$ using the Eulerian approach is:

$$U_E(x,y,t_e) = \sum_0^{t_e} u(x,y,t)\Delta t \tag{19}$$

5 $$V_E(x,y,t_e) = \sum_0^{t_e} v(x,y,t)\Delta t \tag{20}$$

with $u(x,y,t)$ and $v(x,y,t)$ being the components of the velocity vector obtained by image correlation at coordinates $x, y$ and time $t$, and $\Delta t$ is the time interval between correlated images.

### 3.1.1 Benchmark of Eulerian sum

To benchmark the Eulerian sum we employed synthetic images with a static shear zone. The images are $1000 \times 1000$ pixels
10 and include 60,000 particles with a diameter $3\pm1$ pixels. The $u$ component of the velocity is 0, and the $v$ component varies across a shear zone parallel to the y-axis at coordinate $x = 500$ pixels. The shear zone width is 512 pixels and the velocity varies in the shear zone following the error function from 0 on the left to 4 pixels per frame on the right. Figure 3 shows two

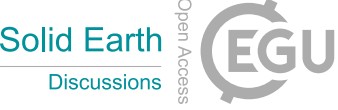

**Figure 8.** PIV results for the moving shear zone synthetic test. The $u$ component of the imposed velocity is homogeneous. The $v$ component changes sign across a shear zone that is advected with $u$. **(a, b, c)**: velocity vectors with the velocity gradient $\Delta v/\Delta x$ at frames 10, 30 and 60. The location of the shear zone moves to the right over time. Vectors are plotted every 4 points in each direction for clarity. **(c, d, e)**: Lagrangian sum of displacements at the same stages plotted as deformed grids. The deformed grid is plotted every 4 points in each direction for clarity. The scalar parameter is the gradient along the x-axis of the cumulative displacement along the y-axis: $\Delta V/\Delta x$. The magnitude of the gradient increases over time, but the shear zone width remains. The red box shows the area plotted in Fig. 12.



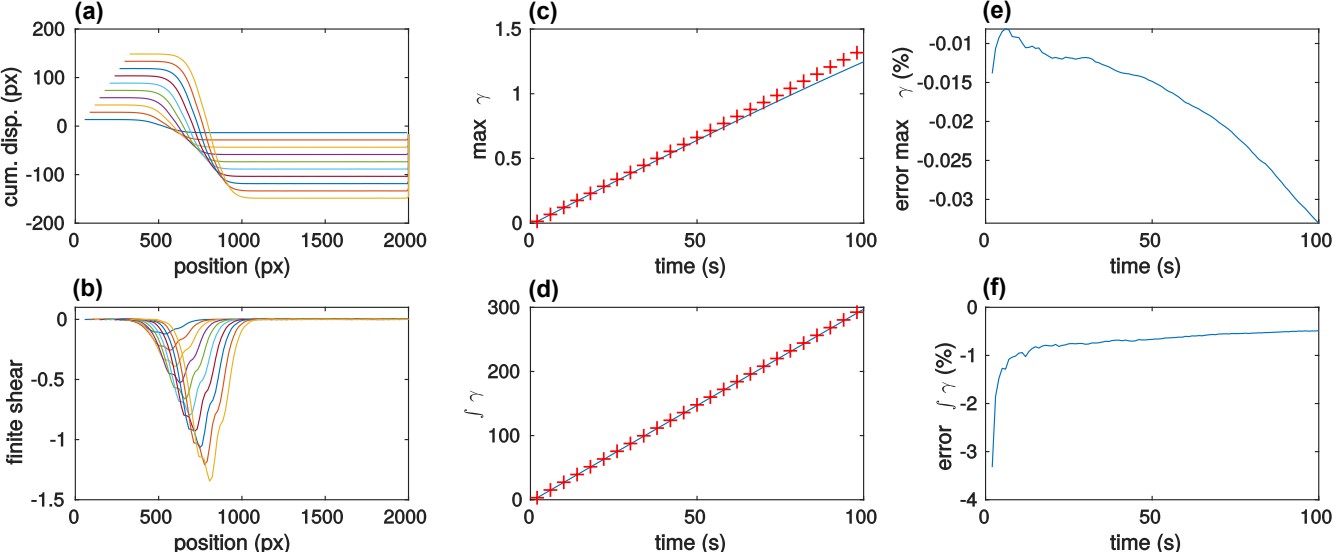

**Figure 9.** Benchmark of the Lagrangian sum of displacements. **(a)**: cumulative displacement profiles across the moving shear zone (profile plotted every 10 frames only for clarity); **(b)**: gradients of the cumulative displacement profiles from panel a; **(c)**: evolution of the peak of the gradient of the cumulative displacements (i.e. max shear, $\gamma$) versus time (assuming 1 s between frames in synthetic tests) measured from PIV (red crosses) versus the solution (blue line) every 4 frames; **(d)**: integral of the shear profile versus time measured from PIV (red crosses) and solution (blue line) every 4 frames; **(e)**: error on the peak cumulative shear strain; **(f)**: error on the integrated cumulative shear strain.

stages of the benchmark test at the beginning and end of the simulation. Figure 7a shows profiles of the Eulerian cumulative displacement across the shear zone every 5 frames during the simulation. Figure 7b shows the gradient of the cumulative displacement along the same profiles (i.e. cumulative shear strain). It can be seen that the shear zone remains static but the amount of accommodated shear increases over time. We compared the peak value of the cumulative shear in the profile to the value derived from the velocity field employed to advect the particle (Fig. 7c). We also compared the integrated cumulative shear in the profile to the values derived from the velocity field employed to advect the particles. Both peak value of the cumulative shear and the integrated cumulative shear increase linearly with rates very similar to the expected behaviour. We quantified the difference between the values derived from the PIV analysis and the solution. For both the peak cumulative shear and the integrated cumulative shear the error is largest at the beginning of the sum, and reduces to about 2% for the peak shear strain and less than 0.5% for the integrated cumulative shear strain. We conclude that the Eulerian sum provides a very reliable way to estimate the amount of shear accommodated in the static shear zone.

## 3.2 Lagrangian sum of displacements

The Lagrangian sum requires that the incremental displacements, which are summed up, be advected with the flow. This poses some difficulties because the displacement is defined at each time step only on the points of the vector grid but not in between.





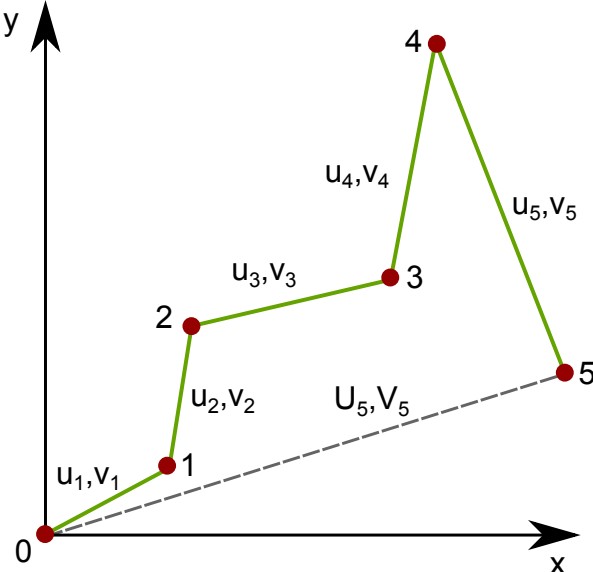

**Figure 10.** Illustration of incremental and cumulative Lagrangian displacements and track length. Material point (red dot) moves from positions 0 to 5. The velocity vectors at time step $i$ are $u_i, v_i$. The cumulative displacements are $U_i, V_i$ (dashed line). The track length is the length of the path: the length of the green line.

However, the incremental displacement (i.e., between successive images) is generally significantly smaller than the vector grid resolution. For example, the vector grid may have points every 32 pixels but the incremental displacement maybe up to a few pixels only. The vectors that are summed up are thus vectors, which have been interpolated from neighbouring points on the vector grid.

5    Here, the value at each requested point that falls in between known vector points is calculated using the three nearest non-collinear neighbours. If the three nearest neighbours method fails because the points are collinear, the value of the unsampled point is calculated as the weighted average of the known values of the $N$ nearest neighbours. The weights of the neighbours are inversely related to the distances between the prediction location and the sampled locations. This method allows rapid calculation of the Lagrangian sum of displacements even for large grids.

10    The components $U_L, V_L$ of the finite displacement of a material point $a$ at time $t_e$ is:

$$U_L(a, t_e) = \sum_0^{t_e} u(X(a,t))\Delta t \tag{21}$$

$$V_L(a, t_e) = \sum_0^{t_e} v(X(a,t))\Delta t \tag{22}$$

where $X(a,t)$ is the location of the material point $a$ at time $t$, and $u(X(a,t))$ is the interpolated velocity component of the material point $a$ at location $X(a,t)$.




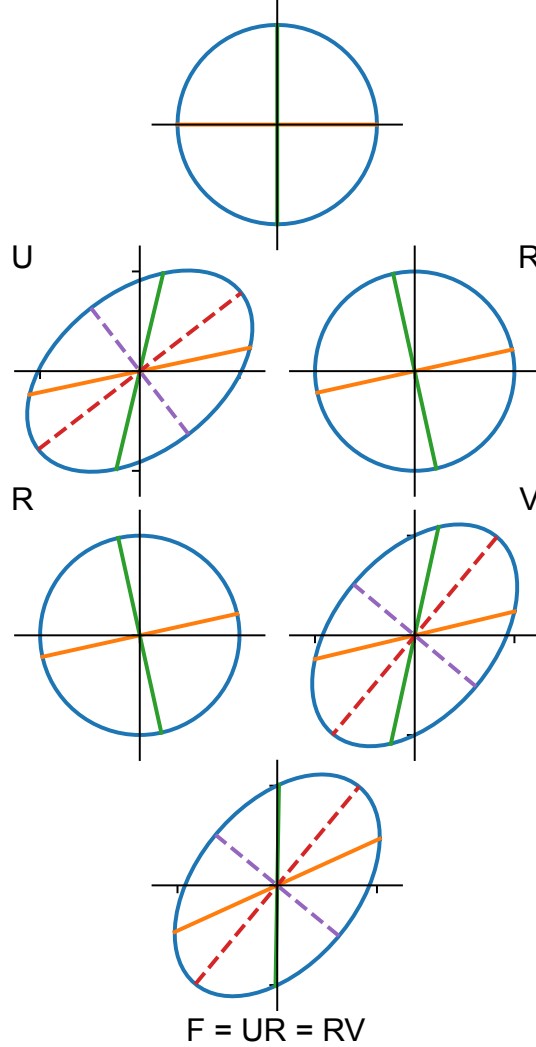

**Figure 11.** Polar decomposition of the deformation gradient tensor **F** into a finite rigid-body rotation tensor **R** and either a left-stretch tensor **V** or right-stretch tensor **U**. Note that **F** and **V** have the same orientations of their principal components. The green and orange lines are initially parallel to the axes of the Cartesian reference frame and here illustrate the rotation of the unit circle (top). The dashed lines are the long and short axes of the deformation ellipsoids.

### 3.2.1 Benchmark of Lagrangian sum

To benchmark the Lagrangian sum, we created a series of synthetic images were the particles are moved homogeneously along the x axis, but heterogeneously along the y axis. On the left-hand side of a shear zone parallel to the y axis, the particles move downward, while on the right-hand side of the shear zone the particles move up. In between is a shear zone where the y velocity component varies along x as an error function. The shear profile across the shear zone is thus a bell-shaped curve. At each time




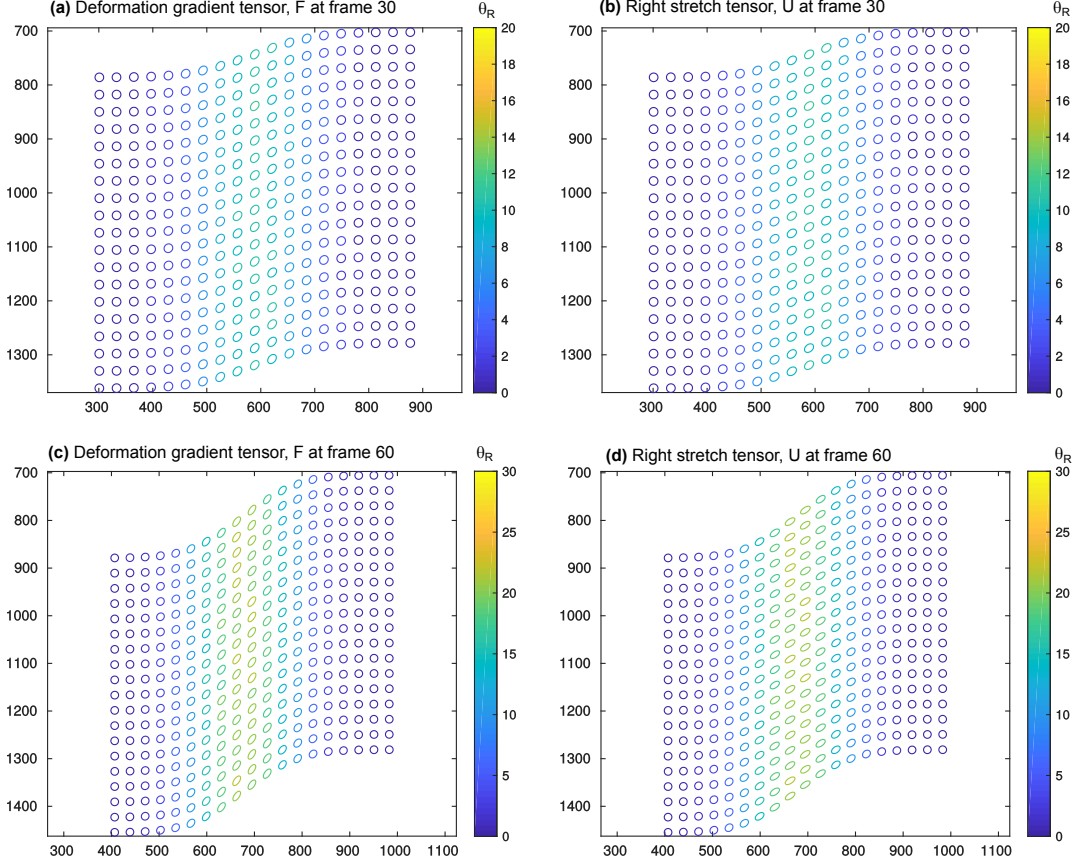

**Figure 12.** Deformation gradient tensor $\mathbf{F}$ and right-stretch tensor $\mathbf{U}$ calculated from the Lagrangian sum of displacements at frame 30 and 60 of the moving shear zone synthetic test. For clarity, only one in four points are plotted as ellipses. The colour indicates the amount of finite rigid-body rotation obtained via tensor $\mathbf{R}$ (in degrees counterclockwise). Positions of the plotted areas are indicated in Fig. 8.

step, the x position of the shear zone is advected along the x axis with the particle. An Eulerian sum would thus smear the shear zone over the various positions it has in the recording, but a Lagrangian sum should maintain the width of the shear zone. The images included 240,000 particles with a diameter of $3\pm1$ pixels. The width of the shear zone is 512 pixels, the horizontal incremental displacement is 3 pixels, while the y-axis velocity component varies between $-1.5$ and $1.5$ pixels. We calculated

5    the incremental displacements (Fig. 8a-c) and summed using the Lagrangian mode (Fig. 8d-f). Figure 9a shows the profiles of $v$ across the shear zone at successive steps, while figure 9b shows the gradient of the finite displacement component $V$. It can be seen that the peak is shifting appropriately, and the width of the shear zone appears to be maintained properly. To quantify further, we calculated the peak of the shear profile and its integral at each time step and compared them with values computed from the equation employed to advect the particles in the synthetic images (Fig. 9c-d). The error on the magnitude of the peak

10   increases over time, but is less than 0.1% after 100 steps, corresponding to a relative displacement along y of 300 pixels. The





error on the integrated shear decreases rapidly in the first steps and then continues to slowly decrease until step 100. The final error on the integrated shear strain is less than 1%.

We conclude that the calculation of the Lagrangian sum is precise when the incremental displacement can be measured accurately. Of course if the quality of the experimental images does not permit a precise measure of the incremental displace-

ments, the finite displacements will also reflect this inaccuracy. However, the summation process is itself accurate. We note here that we employed a simple linear interpolation from the three-nearest neighbours to calculate the interpolated velocity and incremental displacement that are summed up. This interpolation technique was chosen for its rapidity and ability to get values at precisely the unsampled point. It is possible that a spline or cubic polynomial employing more neighbouring known points might provide an even better accuracy.

### 3.2.2   Track length

The finite strain tensor is calculated using the Lagrangian cumulative displacement above and therefore compares the initial and final states irrespective of the strain path (Fig. 10). A measure of the length of this path is provided with the following parameter:

$$L(a,t_e) = \sum_{0}^{t_e} \Delta t \sqrt{u(a,t)^2 + v(a,t)^2} \qquad (23)$$

The components of the incremental displacement of a material point $a$ are interpolated and employed to calculate the magnitude of the incremental displacement, which is summed over the duration of the recording.

### 3.3   Finite strain tensor

The Lagrangian finite strain tensor $\mathbf{E}$ is calculated using the gradients of the Lagrangian cumulative displacements and the initial relative position of the points. In 2D the Lagrangian finite strain tensor is :

$$\mathbf{E} = \begin{bmatrix} E_{xx} & E_{xy} \\ E_{xy} & E_{yy} \end{bmatrix} \qquad (24)$$

with

$$E_{xx} = \frac{1}{2}\left( \frac{\Delta U}{\Delta x} + \frac{\Delta U}{\Delta x} + \left( \frac{\Delta U}{\Delta x}\frac{\Delta U}{\Delta x} + \frac{\Delta V}{\Delta x}\frac{\Delta V}{\Delta x} \right) \right) \qquad (25)$$

$$E_{yy} = \frac{1}{2}\left( \frac{\Delta V}{\Delta y} + \frac{\Delta V}{\Delta y} + \left( \frac{\Delta U}{\Delta y}\frac{\Delta U}{\Delta y} + \frac{\Delta V}{\Delta y}\frac{\Delta V}{\Delta y} \right) \right) \qquad (26)$$

$$E_{xy} = \frac{1}{2}\left( \frac{\Delta U}{\Delta y} + \frac{\Delta V}{\Delta x} + \left( \frac{\Delta U}{\Delta x}\frac{\Delta U}{\Delta y} + \frac{\Delta V}{\Delta x}\frac{\Delta V}{\Delta y} \right) \right) \qquad (27)$$

$$E_{yx} = \frac{1}{2}\left( \frac{\Delta V}{\Delta x} + \frac{\Delta U}{\Delta y} + \left( \frac{\Delta V}{\Delta y}\frac{\Delta V}{\Delta x} + \frac{\Delta U}{\Delta y}\frac{\Delta U}{\Delta x} \right) \right) \qquad (28)$$

It can be seen that $\mathbf{E}$ is symmetrical and contains second-order terms in contrast to the small strain tensor (Eq. 5 XYZ).





### 3.4 Invariants of Lagrangian finite strain tensor

The first and second invariants of the Lagrangian finite strain tensor are:

$$I_{1E} = E_{xx} + E_{yy} \tag{29}$$

$$I_{2E} = E_{xx}E_{yy} - E_{xy}^2 \tag{30}$$

### 3.5 Rotation and stretch tensors

With finite strain the displacement gradient tensor cannot be decomposed additively into symmetric strain and anti-symmetric rotation parts. However, the deformation gradient tensor can be decomposed into the product of two tensors: one accounting for the stretch and one for the rigid-body rotation.

The deformation gradient tensor is defined as the gradient of the deformed distances of material points relative to their initial
distances:

$$F_{ij} = \frac{\Delta X_i}{\Delta x_j} \tag{31}$$

with $\Delta X_i$ the distance between the points in the direction $i$ ($x$ or $y$) in the deformed state, while $\Delta x_j$ is the distance between the points in the direction $j$ in the undeformed state. Since we can extract the position of points in the deformed state using the Lagrangian sum of displacements and already know the initial positions of these points, the deformation gradient tensor can be
extracted from the Lagrangian sum.

The deformation gradient tensor can be decomposed multiplicatively in two ways:

$$\mathbf{F} = \mathbf{R}\mathbf{U} = \mathbf{V}\mathbf{R} \tag{32}$$

where $\mathbf{R}$ is the rotation tensor, $\mathbf{U}$ is the right-stretch tensor, and $\mathbf{V}$ is the left-stretch tensor. In the first case, the rigid-body rotation occurs first followed by stretching, while in the second case stretching occurs before rigid-body rotation. In both case
the final stage is identical (Fig. 11). The right-stretch tensor $\mathbf{U}$ can be obtained from $\mathbf{F}$:

$$\mathbf{U} = (\mathbf{F}^T\mathbf{F})^{1/2} \tag{33}$$

and then the rotation tensor can be obtained from $\mathbf{F}$ and $\mathbf{U}$:

$$\mathbf{R} = \mathbf{F}\mathbf{U}^{-1} \tag{34}$$

In 2-D the rigid body rotation can be characterised by a single rotation angle:

$$\Theta_f = \cos^{-1}(R_{11}) \tag{35}$$

The left-stretch tensor $\mathbf{V}$ can also be obtained from $\mathbf{F}$:

$$\mathbf{V} = (\mathbf{F}\mathbf{F}^T)^{1/2} \tag{36}$$





In our code, the components of **F** are calculated for each Lagrangian point, and then **F** is assembled. Next, **U**, **R** and **V** are calculated. Finally, the magnitudes and orientations of the principal values are calculated for **F** and **U**.

Figure 12 shows the finite strain ellipses using the deformation gradient tensor **F** and the right-stretch tensor **U** at frames 30 and 60 of our synthetic test on a moving shear zone employed to measure the accuracy of the Lagrangian sum (Figs. 8 and 9).

As expected, the ellipsoid becomes more elongated near the centre line of the shear zone, and with time. The amount of finite rigid-body rotation also increases towards the centre line of the shear zone and over time. The measure of stretch before rigid body rotation from **U** shows the principal stretches forming the expected $S_1$ stretch lines (Ramsay and Huber, 1987).

## 4   Conclusions

Particle Imaging Velocimetry of analogue models of tectonics can be performed with inexpensive, high-resolution commercial

DSLR cameras and open-sourced scripts for MATLAB. Our GUI package TecPIV provides a way to perform the calibration and rectification of the model images prior to the correlation of successive image pairs.

A new correlation algorithm has been implemented, allowing the use of the multipass procedure with decreasing window size and window deformation. Synthetic tests confirm that this method improves the ability to capture displacements in and around narrow deformation bands compared to the previous approach using single pass with larger overlap (Boutelier, 2016).

A region of interest is defined as a rectangular box encompassing the deforming model, inside which vectors are calculated. A deforming mask is also calculated for each time step. The mask corresponds to the ROI for the first time step, but is advected with the vectors in subsequent steps.

The software package now provides access to the 2-D small strain tensor components, its invariants as well as the principal extensions, and maximum shear. The incremental deformation ellipsoid, using principal stretches instead of extension, can be

derived everywhere in the imaged model and for each time step.

Two approaches are proposed for the calculation of the finite displacements: Eulerian or Lagrangian sums. The Lagrangian sum may be more appropriate for the analysis of tectonic models as it allows calculating the amount of deformation accommodated by a moving feature. Synthetic tests demonstrate that when the incremental displacements are accurate, the sums are also accurate as only a very small error is introduced due to the interpolation required for the summation process. The Lagrangian

sum of displacements allows the calculation of the 2-D Lagrangian finite strain tensor and its invariants. In addition, the 2-D deformation gradient tensor is calculated from the model deformed state (i.e., from the Lagrangian sum of displacements and initial positions). The 2-D deformation gradient tensor can then be decomposed into the finite rigid-body rotation tensor and either a left- or right- finite stretch tensor. Deformation ellipses can be plotted and the effect of the rigid body-rotations and stretch measured and visualised.

We conclude that the PIV analysis of 2-D, plane-strain, analogue models of tectonics can provide simultaneously the incremental and cumulative strain tensor. The finite deformation was traditionally analysed in analogue models using passive markers (i.e. Graveleau et al., 2012, and reference therein). Particle Imaging Velocimetry technique born of fluid dynamics experiments has recently provided access to incremental, or instantaneous, deformation rates but often neglected finite strain.




The cumulative effects of finite strain are required to permit better comparisons of models to geological field observations. The calculation of the Lagrangian sum allows bringing back the measure of the finite deformation.

*Code and data availability.* The MATLAB scripts for making the synthetic images are available at: https://github.com/davidboutelier/Synthetic_PIV_images. The code for the PIV analysis is available at: https://github.com/davidboutelier/TecPIV.

*Author contributions.* DB wrote the algorithm. DB, CS & KRL wrote the manuscript.

*Competing interests.* No competing interests are present

## Appendix A: Nomenclature

$x$ : coordinate along x-axis

$y$ : coordinate along y-axis

$u$ : component of velocity vector along x axis

$v$ : component of velocity vector along y axis

$\mathbf{v}$ : velocity vector

$\mathbf{s}$ : displacement vector

$\Delta t$ : time between images

$\Delta x$ : resolution of vector grids along x-axis

$\Delta y$ : resolution of vector grids along y-axis

$i, j$ : indices of grids along x and y dimensions

$\mathbf{D}$ : velocity gradient tensor

$\mathbf{S}$ : symmetric part of velocity gradient tensor

$\mathbf{A}$ : anti symmetric part of velocity gradient tensor

$\omega$ : vorticity

$I_{1e}$ : first invariant of small strain tensor

$I_{2e}$ : second invariant of small strain tensor

$\theta_p$ : Orientation of principal strain

$e_{min}$ : Minimum principal strain

$e_{max}$ : Minimum principal strain

$\gamma_{max}$ : Maximum shear

$U_E$ : Component of Eulerian finite displacement $(x)$





$V_E$ : Component of Eulerian finite displacement ($y$)

$U_L$ : Component of Lagrangian finite displacement ($x$)

$V_L$ : Component of Lagrangian finite displacement ($y$)

$L$ : Lagrangian track length

5    $\mathbf{E}$ : Lagrangian finite strain tensor

$\mathbf{F}$ : Deformation gradient tensor

$\mathbf{R}$ : Finite rigid-body rotation tensor

$\mathbf{U}$ : Right stretch tensor

$\mathbf{V}$ : Left stretch tensor

10    $\Theta_f$ : Finite rigid body rotation angle

*Acknowledgements.* The authors acknowledge funding for this study from the Australian Research Council via Discovery Project DP14001200.





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
