# Peer review of "2-D finite displacements and strain from Particle Imaging Velocimetry (PIV) analysis of tectonic analogue models with TecPIV"

_Solid Earth, 2019_

## Referee Comment (RC1) · Matthias Rosenau (Referee) · 10 May 2019

Review of manuscript se-2019-67

The manuscript documents methodological developments and synthetic benchmarks of an open source software for digital image correlation (or particle image velocimetry, PIV) tailored to applications in analogue tectonic modelling (TecPIV by Boutelier (2016)). The developments include multipass processing with decreasing window size and window deformation and the derivation of various incremental and finite strain components in Eulerian and Lagrangian reference systems. Especially summing up incremental strain in a Lagrangian reference frame meets the demands of the commu-

nity, as this is the reference frame of geoscientists' observations at long time scales at which material and structures are advected. These developments make TecPIV competitive with respect to commercial software packages (e.g. LaVision Davis Strain-master). In my opinion it is a laudable effort by the authors to develop, improve and maintain TecPIV and the paper is a highly appreciated contribution to be published in SE. I enjoyed reading the manuscript which is at a very mature stage and needs to my opinion only minor (mainly technical) revisions before publication.

General comments:

Structure and language:

The paper is straight forwardly structured and very well written at a level of detail that allows appreciating the mathematics behind the typically "colourful" results.

Mathematics:

I have to admit not having checked all formulas for their correctness by myself (I would not be able to do it) but from what I could verify I have the impression of things being correctly documented.

Figures:

Figures are all usefull, required and generally well designed.

Specific comments:

Title:

In the title you constrict the use to applications for plane strain. I probably understand your motivation to do so because the method is 2D but in contrast to the synthetic benchmarks (which are truly 2D) any real analogue model (free) surface will deviate at least locally from plane strain. Also because in the main text you never recall the issue of plane strain (not before the conclusions), so I suggest to delete "plane strain" from the title and instead discuss the limitations of 2D analysis shortly elsewhere.

Terms and definitions:

"PIV": "PIV" is an essential acronym which should be defined earlier (title or abstract) than it is currently done (where you define it in the third paragraph of the intro). You may also quickly clarify the use of the term "PIV" in parallel to the emerging preference of the term "DIC" (digital image correlation) for applications outside the fluid dynamic context. In understand you stick to "PIV" because the software has this name and you may not want to change it. Even if this would be the only reason, it should be spoken out frankly.

"TecPIV" first (and only) appears in the conclusions although it is what the paper is about. It should be introduced in the introduction and abstract (if not in the title).

Page 2 Line 3ff:

The reasoning for not using PIV cameras is (my perception) nowadays mainly the costs, not the resolution. I therefore suggest to update the sentence on "relatively low-resolution". You say that PIV cameras have low resolution (<10 MPx), which is strictly valid only for imaging frequencies beyond what is typically used in analogue modelling of tectonic processes and for cameras with a color depth beyond 14 bit. There now exist also higher resolution (up to 29 MPx, 14 bit) PIV cameras which run at frequencies up to a few Hz and I assume with a time lag of a few years they will keep pace with DSLR camera developments in term of spatial resolution. The sensitivity of PIV cameras might indeed be of minor importance for the tectonic analogue modelling community, unless you work with pulsed light or under low light conditions. A short sentence clarifying the role of a proper dynamic range (preferentially at least 12 bit) for image correlation accuracy could be helpful in this context. Lastly, analogue "seismo-tectonic" models, where high rates are an issue, requires imaging frequencies beyond what is possible with consumer-grade cameras at a decent level of image quality.

Technical comments:

Figures:

A) Increase font size in all panels showing the velocity fields (numbers on axes and color bars).

B) Figure 3 & 5: The grey background is not well suited to appreciate the velocity field, especially the low levels. A white zero-level (as actually the color bar indicates) is better. The undeformed areas of the masks are not really visible, maybe add some space outside the frame, make the lines thicker etc.

Criteria:

Does the paper address relevant scientific questions within the scope of SE? YES

Does the paper present novel concepts, ideas, tools, or data? YES

Are substantial conclusions reached? YES (technique-wise)

Are the scientific methods and assumptions valid and clearly outlined? YES

Are the results sufficient to support the interpretations and conclusions? YES

Is the description of experiments and calculations sufficiently complete and precise to allow their reproduction by fellow scientists (traceability of results)? YES

Do the authors give proper credit to related work and clearly indicate their own new/original contribution? YES

Does the title clearly reflect the contents of the paper? Maybe not (see comments above)

Does the abstract provide a concise and complete summary? YES

Is the overall presentation well structured and clear? YES

Is the language fluent and precise? YES

Are mathematical formulae, symbols, abbreviations, and units correctly defined and

used? YES

Should any parts of the paper (text, formulae, figures, tables) be clarified, reduced, combined, or eliminated? NO

Are the number and quality of references appropriate? YES

Is the amount and quality of supplementary material appropriate? n.a.

_________ Matthias Rosenau

---

## Referee Comment (RC2) · Michele Cooke (Referee) · 21 May 2019

The paper presents a new tool that provides critical information on finite strain field evoution within scaled analog experiments of tectonic processes. The tool provides information about Lagrangian strain fields that is not easily available with other open-access tools. The authors validate the approach within several simple but illustrative examples that demonstrate the power of the approach. This is a tool that I look forward to using with my experimental data!

I recommend that you use the name of the software, tecPIV, throughout the paper to help the reader associate the innovative technique with the specific code that you've

developed.

Digital Image Correlation with Lagrangian reference frame have been used with analog models in the past and it may be helpful to include some of these references. For example, Tonenboehn et al., (2018) used Particle Tracking Velocimetry (PTV) to track advection through restraining bends. The PTV results are not as useful as the finite strain presented because they only produce displacement paths, rather than full strain field. We also found that the PTV works best with a different type and spacing of markers than the PIV; thus, requiring two different experiments to get both sets of data. Pointing out this deficiency of PTV could provide an opportunity to highlight further advantages of tecPIV.

Would there be a CPU benefit to adaptive refinement, where third and fourth pass finer resolution interrogations were applied to areas with changing displacement and higher while areas of rigid translation just had 1 or 2? This would be similar to the adaptive remeshing that is employed in some finite element method models. The resulting data would not be on a regular grid so the CPU benefit might have to be weighed with the awkwardness of the non-gridded result. The discussion of the paper could outline the utility of this.

Presentation of the standard equations is helpful though much of the text as it allows the reader to follow the principles of the analysis. One exception is that the equations for calculating principal strain become a bit pedantic. Because these can be found in any mechanics or structural geology textbook, equations 12-15 could be removed for brevity and standard textbook can be cited. Citations to standard textbooks would be helpful throughout. For example, section 3.5 is new to me and I had a hard time appreciating the reason to set up the strain tensors in either left or right stretch. Citations of a textbook or two would give me resources to better appreciate this approach.

Section 2.7 should be called the incremental principal strain. The maximum shear orientation here is noted to be 45ËŽ degrees from the principal strain orientations but

I believe this is only true for incremental strain where vorticity is near zero.

I greatly enjoyed reading about the Eularian sum approach. I tried to code this up myself at one point and the accumulated errors in the summation were horrible. One reason for this was that my strain field was not static and so I should have used Lagrangian, which I eventual did using PTVLab (Toeneboehn et al, 2018). This paper could be more up front in its recommendations to readers on when to use Eularian and when to use Lagrangian. For example, Figure 10 is helpful for delineating the incremental and cumulative displacements. I wonder if adding a grid of points (vector grid) to this figure would help demonstrate why an Eularian summation is not the best approach for this problem.

The benchmarks for testing the Eularian analysis and Lagrangian summation are very well done. Because the tests are synthetic, they report a minimum error for the analysis. This is mentioned on page 18 line 5 and got me thinking about imaging issues. It would be interesting to see how the same tests perform with random noise added to the velocities. This could simulate the potential impact various experimental effects such as slightly out of focus cameras, unclear resolution of individual particles etc. For example, it would be good to know if the technique amplifies errors inherent aleatoric uncertainties or if these errors are just passed through the analysis without amplification.

Specific comments: Page 1 line 21 'pass through an evolution' ← awkward Page 6 line 36 "..less unique distribution of ??? values than a large one." Are this displacement values, image correlation values or something else? Page 7 line 7: narrow shear zones. (plural) Page 7 line 10: . . . distribution of image values . . .. Page 7 line 14:.. where models produce deformation (or rigid body rotation) is to calculate. . . Page 8 line 2: The change of coordinate system doesn't have to be associated wit rotation. One could arbitrarily assign a different coordinate system. Page 12 line 1: .. were ← should be where Page 16 line 4: .. were ← should be where Page 18 line 11:. Sentence is confusing and could be refined for clarity. Above? Section 3.4 The invariants are

the same for Eularian and Lagrangian so don't need to repeat these equations. This section can be removed. Page 19 line 6: comma after strain Figure 11 could use more guidance for readers unfamiliar with the approach. Numbering of the deformation can show which is first and which is second. Maybe set up as XXX + YYY = ZZZ For the two cases and then the reader can see that the result is the same for the two cases. Page 20 line 14: Deformation zones (deformation bands are a particular structure and the technique here can be applied more broadly than just to deformation bands.)

---

## Author Comment (AC1) · 18 Jun 2019

We thank the reviewer for the constructive comments on the manuscript. We have taken these comments into account to produce a significantly clearer manuscript.

Below are the specific comments and how they have been addressed in the revision.

COMMENT: "PIV": "PIV" is an essential acronym which should be defined earlier (title or abstract) than it is currently done (where you define it in the third paragraph of the intro). You may also quickly clarify the use of the term "PIV" in parallel to the emerging preference of the term "DIC" (digital image correlation) for applications outside the fluid

dynamic context. In understand you stick to "PIV" because the software has this name and you may not want to change it. Even if this would be the only reason, it should be spoken out frankly.

RESPONSE: We have now included the term PIV and what it means in the title as suggested by the reviewer.

COMMENT: "TecPIV" first (and only) appears in the conclusions although it is what the paper is about. It should be introduced in the introduction and abstract (if not in the title). Page 2 Line 3ff:

RESPONSE: We now mention TecPIV earlier and multiple times throughout the manuscript.

COMMENT: The reasoning for not using PIV cameras is (my perception) nowadays mainly the costs, not the resolution. I therefore suggest to update the sentence on "relatively low-resolution". You say that PIV cameras have low resolution (<10 MPx), which is strictly valid only for imaging frequencies beyond what is typically used in analogue modelling of tectonic processes and for cameras with a color depth beyond 14 bit. There now exist also higher resolution (up to 29 MPx, 14 bit) PIV cameras which run at frequencies up to a few Hz and I assume with a time lag of a few years they will keep pace with DSLR camera developments in term of spatial resolution. The sensitivity of PIV cameras might indeed be of minor importance for the tectonic analogue modelling community, unless you work with pulsed light or under low light conditions. A short sentence clarifying the role of a proper dynamic range (preferentially at least 12 bit) for image correlation accuracy could be helpful in this context. Lastly, analogue "seismo-tectonic" models, where high rates are an issue, requires imaging frequencies beyond what is possible with consumer-grade cameras at a decent level of image quality.

RESPONSE: We expanded this section to include the valid point made by the reviewer.

COMMENT: Figures: A) Increase font size in all panels showing the velocity fields (numbers on axes and color bars).

RESPONSE: We have increased the font size of all panels and most figures. The labels are all font 7 or larger.

COMMENT: Figures B) Figure 3 & 5: The grey background is not well suited to appreciate the velocity field, especially the low levels. A white zero-level (as actually the color bar indicates) is better. The undeformed areas of the masks are not really visible, maybe add some space outside the frame, make the lines thicker etc.

RESPONSE: This is because the derivative of the velocity, which is plotted in colour, is overlaying the view of the model from which it derives. This may not be relevant for the synthetic images employed in this study but is rather important in multiple models. To blend the colour image of the derivative and the view of the model surface, the derivative is made semi-transparent. This is why the white colour does not appear so white when plotted on top of black. We believe this small compromise brings significant value.

COMMENT: Title: In the title you constrict the use to applications for plane strain. I probably understand your motivation to do so because the method is 2D but in contrast to the synthetic benchmarks (which are truly 2D) any real analogue model (free) surface will deviate at least locally from plane strain. Also because in the main text you never recall the issue of plane strain (not before the conclusions), so I suggest to delete "plane strain" from the title and instead discuss the limitations of 2D analysis shortly elsewhere.

RESPONSE: We changed the title following the reviewer's advice.
* * *

---

## Author Comment (AC2) · 18 Jun 2019

We thank the reviewer for the constructive comments on the manuscript. We havetaken these comments into account to produce a significantly clearer manuscript. Below are the specific comments and how they have been addressed in the revision.

COMMENT: I recommend that you use the name of the software, tecPIV, throughout the paper to help the reader associate the innovative technique with the specific code that you've developed.

RESPONSE: The name now appears multiple times throughout the paper.

[Figure]

COMMENT: Digital Image Correlation with Lagrangian reference frame have been used with analog models in the past and it may be helpful to include some of these references. For example, Tonenboehn et al., (2018) used Particle Tracking Velocimetry (PTV) to track advection through restraining bends. The PTV results are not as useful as the finite strain presented because they only produce displacement paths, rather than full strain field. We also found that the PTV works best with a different type and spacing of markers than the PIV; thus, requiring two different experiments to get both sets of data. Pointing out this deficiency of PTV could provide an opportunity to highlight further advantages of tecPIV.

RESPONSE: The introduction now mentions the alternative use of the PTV and its limitation. In addition, the discussion section also brings back the idea of PTV, in an ultimate step after PIV in future developments.

COMMENT: Would there be a CPU benefit to adaptive refinement, where third and fourth pass finer resolution interrogations were applied to areas with changing displacement and higher while areas of rigid translation just had 1 or 2? This would be similar to the adaptive remeshing that is employed in some finite element method models. The resulting data would not be on a regular grid so the CPU benefit might have to be weighed with the awkwardness of the non-gridded result. The discussion of the paper could outline the utility of this.

RESPONSE: This proposition has been added to a new discussion section about future development. It is a challenging but interesting idea.

COMMENT: Presentation of the standard equations is helpful though much of the text as it allows the reader to follow the principles of the analysis. One exception is that the equations for calculating principal strain become a bit pedantic. Because these can be found in any mechanics or structural geology textbook, equations 12-15 could be removed for brevity and standard textbook can be cited.

RESPONSE: We have followed the recommendation and removed Eqs. 12 to 14. Eq.

15 is required to explain how the orientation of the principal strain is calculated as well as the magnitude.

COMMENT: Citations to standard textbooks would be helpful throughout. For example, section 3.5 is new to me and I had a hard time appreciating the reason to set up the strain tensors in either left or right stretch. Citations of a textbook or two would give me resources to better appreciate this approach.

RESPOENSE: We added a reference to a textbook as suggested.

COMMENT: Section 2.7 should be called the incremental principal strain. The maximum shear orientation here is noted to be 45ЁŽ degrees from the principal strain orientations but I believe this is only true for incremental strain where vorticity is near zero.

RESPONSE: Yes this section is about incremental strain. This is now clearly indicated in the section title.

COMMENT: I greatly enjoyed reading about the Eularian sum approach. I tried to code this up myself at one point and the accumulated errors in the summation were horrible. One reason for this was that my strain field was not static and so I should have used La- grangian, which I eventual did using PTVLab (Toeneboehn et al, 2018). This paper could be more up front in its recommendations to readers on when to use Eularian and when to use Lagrangian.

RESPONSE: A section has been developed about the advantages and disadvantages of both methods and when they are more appropriate. Whether the strain field is static or not is a key factor to take into account.

COMMENT: For example, Figure 10 is helpful for delineating the in- cremental and cumulative displacements. I wonder if adding a grid of points (vector grid) to this figure would help demonstrate why an Eularian summation is not the best approach for this problem.

RESPONSE: Figure 10 is designed to show that the finite displacements do not account for the length of the path. We believe tracking a single material point conveys this specific message best.

COMMENT: The benchmarks for testing the Eularian analysis and Lagrangian summation are very well done. Because the tests are synthetic, they report a minimum error for the anal- ysis. This is mentioned on page 18 line 5 and got me thinking about imaging issues. It would be interesting to see how the same tests perform with random noise added to the velocities. This could simulate the potential impact various experimental effects such as slightly out of focus cameras, unclear resolution of individual particles etc. For example, it would be good to know if the technique amplifies errors inherent aleatoric uncertainties or if these errors are just passed through the analysis without amplifica- tion.

RESPONSE: We focused on the ability of the methods to produce an accurate sum without adding errors. It has been demonstrated that random noise or error in the incremental data vanishes in the cumulative data because only the signal is constant. Schrank et al. 2008 used this property with analogue shear zone where the incremental signal was very small but the changes in trends where noticeable in the stronger cumulative signal.

Specific comments: COMMENT: Page 1 line 21 'pass through an evolution' ← awkward

RESPONSE: Sentence has been modified

COMMENT: Page 6 line 36 "..less unique distribution of ??? values than a large one." Are this displacement values, image correlation values or something else?

RESPONSE: Intensity values are correlated. This is now clarified.

COMMENT: Page 7 line 7: narrow shear zones. (plural)

RESPONSE: This has been corrected

COMMENT: Page 7 line 10: . . . distribution of image values . . ..

RESPONSE: This has been corrected

COMMENT: Page 7 line 14:..where models produce deformation (or rigid body rotation) is to calculate. . .

RESPONSE: This has been corrected

COMMENT: Page 8 line 2: The change of coordinate system doesn't have to be associated wit rotation. One could arbitrarily assign a different coordinate system.

RESPONSE: A translation does not change the deformation tensor, but a rotation will.

COMMENT: Page 12 line 1: .. were ← should be where

RESPONSE: This has been corrected

COMMENT: Page 16 line 4: .. were ← should be where

RESPONSE: This has been corrected

COMMENT: Page 18 line 11:. Sentence is confusing and could be refined for clarity. Above?

RESPONSE: The sentence has been clarified.

COMMENT: Section 3.4 The invariants are the same for Eularian and Lagrangian so don't need to repeat these equations. This section can be removed.

RESPONSE: This has been corrected

COMMENT: Page 19 line 6: comma after strain

RESPONSE: This has been corrected

COMMENT: Figure 11 could use more guidance for readers unfamiliar with the approach. Numbering of the deformation can show which is first and which is second.

[Figure]

Maybe set up as XXX + YYY = ZZZ For the two cases and then the reader can see that the result is the same for the two cases.

RESPONSE: This has been corrected

COMMENT: Page 20 line 14: Deformation zones (deformation bands are a particular structure and the technique here can be applied more broadly than just to deformation bands.)

RESPONSE: This has been corrected

---

## Author Response (AR1)

Dear Editor,

The two reviewers provided simple constructive comments on our manuscript (se-2019-67) which we have taken into account to clarify the document. We have now produced a revision version.

To demonstrate that we have addressed all the comments by the reviewers have pasted in this document their comments and our answers. You will see that most of the comments are simple queries that have been accepted and improved the manuscript.

Respectfully,
David Boutelier

Review of manuscript se-2019-67
The manuscript documents methodological developments and synthetic benchmarks of an open source software for digital image correlation (or particle image velocime-try, PIV) tailored to applications in analogue tectonic modelling (TecPIV by Boutelier(2016)). The developments include multipass processing with decreasing window sizeand window deformation and the derivation of various incremental and finite strain components in Eulerian and Lagrangian reference systems. Especially summing upincremental strain in a Lagrangian reference frame meets the demands of the commu-nity, as this is the reference frame of geoscientists' observations at long time scales at which material and structures are advected. These developments make TecPIV competitive with respect to commercial software packages (e.g. LaVision Davis Strain-master). In my opinion it is a laudable effort by the authors to develop, improve and maintain TecPIV and the paper is a highly appreciated contribution to be published in SE. I enjoyed reading the manuscript which is at a very mature stage and needs to my opinion only minor (mainly technical) revisions before publication.

General comments:
Structure and language:
The paper is straight forwardly structured and very well written at a level of detail that allows appreciating the mathematics behind the typically "colourful" results.
Mathematics:
I have to admit not having checked all formulas for their correctness by myself (I would not be able to do it) but from what I could verify I have the impression of things being correctly documented.
Figures:
Figures are all usefull, required and generally well designed.
Specific comments:

Terms and definitions:
"PIV": "PIV" is an essential acronym which should be defined earlier (title or abstract) than it is currently done (where you define it in the third paragraph of the intro). You may also quickly clarify the use of the term "PIV" in parallel to the emerging preference of the term "DIC" (digital image correlation) for applications outside the fluid dynamic context. In understand you stick to "PIV" because the software has this name and you may not want to change it. Even if this would be the only reason, it should be spoken out frankly.

We have now included the term PIV and what it means in the title as suggested by the reviewer.

"TecPIV" first (and only) appears in the conclusions although it is what the paper is about. It should be introduced in the introduction and abstract (if not in the title).
Page 2 Line 3ff:

We now mention TecPIV earlier and multiple times throughout the manuscript.

The reasoning for not using PIV cameras is (my perception) nowadays mainly the costs, not the resolution. I therefore suggest to update the sentence on "relatively low-resolution". You say that PIV cameras have low resolution (<10 MPx), which is strictly valid only for imaging frequencies beyond what is typically used in analogue modelling of tectonic processes and for cameras with a color depth beyond 14 bit. There now exist also higher resolution (up to 29 MPx, 14 bit) PIV cameras which run at frequencies up to a few Hz and I assume with a time lag of a few years they will keep pace with DSLR camera developments in term of spatial resolution. The sensitivity of PIV cameras might indeed be of minor importance for the tectonic analogue modelling community, unless you work with pulsed light or under low light conditions. A short sentence clarifying the role of a proper dynamic range (preferentially at least 12 bit) for image correlation accuracy could be helpful in this context. Lastly, analogue "seismo-tectonic" models, where high rates are an issue, requires imaging frequencies beyond what is possible with consumer-grade cameras at a decent level of image quality.

We expanded this section to include the valid point made by the reviewer.
Technical comments:

Figures:
A) Increase font size in all panels showing the velocity fields (numbers on axes and color bars).

We have increased the font size of all panels and most figures. The labels are all font 7 or larger.

B) Figure 3 & 5: The grey background is not well suited to appreciate the velocity field, especially the low levels. A white zero-level (as actually the color bar indicates) is better. The undeformed areas of the masks are not really visible, maybe add some space outside the frame, make the lines thicker etc.

This is because the derivative of the velocity, which is plotted in colour, is overlaying the view of the model from which it derives. This may not be relevant for the synthetic images employed in this study but is rather important in multiple models. To blend the colour image of the derivative and the view of the model surface, the derivative is made semi-transparent. This is why the white colour does not appear so white when plotted on top of black. We believe this small compromise brings significant value.

Title:
In the title you constrict the use to applications for plane strain. I probably understand your motivation to do so because the method is 2D but in contrast to the synthetic benchmarks (which are truly 2D) any real analogue model (free) surface will deviate at least locally from plane strain. Also because in the main text you never recall the issue of plane strain (not before the conclusions), so I suggest to delete "plane strain" from the title and instead discuss the limitations of 2D analysis shortly elsewhere.

We changed the title following the reviewer's advice.

Michele Cooke (Referee)

The paper presents a new tool that provides critical information on finite strain field evoution within scaled analog experiments of tectonic processes. The tool provides

information about Lagrangian strain fields that is not easily available with other open-access tools. The authors validate the approach within several simple but illustrative examples that demonstrate the power of the approach. This is a tool that I look forward to using with my experimental data!

I recommend that you use the name of the software, tecPIV, throughout the paper to help the reader associate the innovative technique with the specific code that you've developed.

The name now appears multiple times throughout the paper.

Digital Image Correlation with Lagrangian reference frame have been used with analog models in the past and it may be helpful to include some of these references. For example, Tonenboehn et al., (2018) used Particle Tracking Velocimetry (PTV) to track advection through restraining bends. The PTV results are not as useful as the finite strain presented because they only produce displacement paths, rather than full strain field. We also found that the PTV works best with a different type and spacing of markers than the PIV; thus, requiring two different experiments to get both sets of data. Pointing out this deficiency of PTV could provide an opportunity to highlight further advantages of tecPIV.

The introduction now mentions the alternative use of the PTV and its limitation. In addition, the discussion section also brings back the idea of PTV, in an ultimate step after PIV in future developments.

Would there be a CPU benefit to adaptive refinement, where third and fourth pass finer resolution interrogations were applied to areas with changing displacement and higher while areas of rigid translation just had 1 or 2? This would be similar to the adaptive remeshing that is employed in some finite element method models. The resulting data would not be on a regular grid so the CPU benefit might have to be weighed with the awkwardness of the non-gridded result. The discussion of the paper could outline the utility of this.

This proposition has been added to a new discussion section about future development. It is a challenging but interesting idea.

Presentation of the standard equations is helpful though much of the text as it allows the reader to follow the principles of the analysis. One exception is that the equations for calculating principal strain become a bit pedantic. Because these can be found in any mechanics or structural geology textbook, equations 12-15 could be removed for brevity and standard textbook can be cited.

We have followed the recommendation and removed Eqs. 12 to 14. Eq. 15 is required to explain how the orientation of the principal strain is calculated as well as the magnitude.

Citations to standard textbooks would be helpful throughout. For example, section 3.5 is new to me and I had a hard time appreciating the reason to set up the strain tensors in either left or right stretch. Citations of a textbook or two would give me resources to better appreciate this approach. We added a reference to a textbook as suggested.

Section 2.7 should be called the incremental principal strain. The maximum shear orientation here is noted to be 45ËŽ degrees from the principal strain orientations but I believe this is only true for incremental strain where vorticity is near zero.

Yes this section is about incremental strain. This is now clearly indicated in the section title.

I greatly enjoyed reading about the Eularian sum approach. I tried to code this up myself at one point and the accumulated errors in the summation were horrible. One reason for this was that my strain field was not static and so I should have used Lagrangian, which I eventual did using PTVLab (Toeneboehn et al, 2018). This paper could be more up front in its recommendations to readers on when to use Eularian and when to use Lagrangian.

A section has been developed about the advantages and disadvantages of both methods and when they are more appropriate. Whether the strain field is static or not is a key factor to take into account.

For example, Figure 10 is helpful for delineating the incremental and cumulative displacements. I wonder if adding a grid of points (vector grid) to this figure would help demonstrate why an Eularian summation is not the best approach for this problem.

Figure 10 is designed to show that the finite displacements do not account for the length of the path. We believe tracking a single material point conveys this specific message best.

SED
Interactive
comment
The benchmarks for testing the Eularian analysis and Lagrangian summation are very well done. Because the tests are synthetic, they report a minimum error for the analysis. This is mentioned on page 18 line 5 and got me thinking about imaging issues. It would be interesting to see how the same tests perform with random noise added to the velocities. This could simulate the potential impact various experimental effects such as slightly out of focus cameras, unclear resolution of individual particles etc. For example, it would be good to know if the technique amplifies errors inherent aleatoric uncertainties or if these errors are just passed through the analysis without amplification.

We focused on the ability of the methods to produce an accurate sum without adding errors. It has been demonstrated that random noise or error in the incremental data vanishes in the cumulative data because only the signal is constant. Schrank et al. 2008 used this property with analogue shear zone where the incremental signal was very small but the changes in trends where noticeable in the stronger cumulative signal.

Specific comments:
Page 1 line 21 'pass through an evolution' ← awkward
Sentence has been modified

Page 6 line 36 "..less unique distribution of ??? values than a large one." Are this displacement values, image correlation values or something else?
Intensity values are correlated. This is now clarified.

Page 7 line 7: narrow shear zones. (plural)
This has been corrected

Page 7 line 10: . . . distribution of image values . . ..
This has been corrected

Page 7 line 14:..where models produce deformation (or rigid body rotation) is to calculate. . .
This has been corrected

Page 8 line 2: The change of coordinate system doesn't have to be associated wit rotation. One could arbitrarily assign a different coordinate system.
A translation does not change the deformation tensor, but a rotation will.

Page 12 line 1: .. were ← should be where
This has been corrected

Page 16 line 4: .. were ← should be where
This has been corrected

Page 18 line 11:. Sentence is confusing and could be refined for clarity. Above?
This sentence has been clarified

Section 3.4 The invariants are the same for Eularian and Lagrangian so don't need to repeat these equations. This section can be removed.
This has been corrected

Page 19 line 6: comma after strain
This has been corrected

Figure 11 could use more guidance for readers unfamiliar with the approach. Numbering of the deformation can show which is first and which is second. Maybe set up as XXX + YYY = ZZZ For the two cases and then the reader can see that the result is the same for the two cases.
This has been corrected

Page 20 line 14: Deformation zones (deformation bands are a particular structure and the technique here can be applied more broadly than just to deformation bands.)
This has been corrected